# Anticancer Activity of the Choline Kinase Inhibitor PL48 Is Due to Selective Disruption of Choline Metabolism and Transport Systems in Cancer Cell Lines

**DOI:** 10.3390/pharmaceutics14020426

**Published:** 2022-02-16

**Authors:** Pablo García-Molina, Alberto Sola-Leyva, Pilar M. Luque-Navarro, Alejandro Laso, Pablo Ríos-Marco, Antonio Ríos, Daniela Lanari, Archimede Torretta, Emilio Parisini, Luisa C. López-Cara, Carmen Marco, María P. Carrasco-Jiménez

**Affiliations:** 1Department of Biochemistry and Molecular Biology I, University of Granada, 18071 Granada, Spain; pablo98@correo.ugr.es (P.G.-M.); albertosola@ugr.es (A.S.-L.); alejandrolaso@correo.ugr.es (A.L.); priosm@ugr.es (P.R.-M.); 2Department of Pharmaceutical and Organic Chemistry, University of Granada, 18071 Granada, Spain; pilarluque@ugr.es; 3Department of Pharmaceutical Sciences, University of Perugia, 06123 Perugia, Italy; daniela.lanari@unipg.it; 4Department of Cell Biology, University of Granada, 18071 Granada, Spain; arios@ugr.es; 5Center for Nano Science and Technology @Polimi, Istituto Italiano di Tecnologia, Via Pascoli 70/3, 20133 Milano, Italy; archimede.torretta@iit.it (A.T.); emilio.parisini@osi.lv (E.P.); 6Department of Biotechnology, Latvian Institute of Organic Synthesis, Aizkraukles 21, LV-1006 Riga, Latvia

**Keywords:** cancer, lipid metabolism, choline kinase inhibitors, choline uptake

## Abstract

A large number of different types of cancer have been shown to be associated with an abnormal metabolism of phosphatidylcholine (PC), the main component of eukaryotic cell membranes. Indeed, the overexpression of choline kinase α1 (ChoKα1), the enzyme that catalyses the bioconversion of choline to phosphocholine (PCho), has been found to associate with cell proliferation, oncogenic transformation and carcinogenesis. Hence, ChoKα1 has been described as a possible cancer therapeutic target. Moreover, the choline transporter CTL1 has been shown to be highly expressed in several tumour cell lines. In the present work, we evaluate the antiproliferative effect of PL48, a rationally designed inhibitor of ChoKα1, in MCF7 and HepG2 cell lines. In addition, we illustrate that the predominant mechanism of cellular choline uptake in these cells is mediated by the CTL1 choline transporter. A possible correlation between the inhibition of both choline uptake and ChoKα1 activity and cell proliferation in cancer cell lines is also highlighted. We conclude that the efficacy of this inhibitor on cell proliferation in both cell lines is closely correlated with its capability to block choline uptake and ChoKα1 activity, making both proteins potential targets in cancer therapy.

## 1. Introduction

Today, cancer continues to be one of the main causes of death worldwide, with an enormous social, economic and public health cost. The number of patients affected by cancer has been increasing in recent decades. According to annual data provided by the Global Cancer Observatory, in 2020, there were around 19.3 million cancer cases and approximately 10 million cancer deaths worldwide [1]. For this reason, the development of new effective and translational therapies, such as targeted therapies, is required. In these approaches, specific agents such as low molecular weight inhibitors or antibodies are designed to interact with specific enzymes or proteins that may be involved in important metabolic and/or oncogenic signalling pathways [2]. In this sense, the potential of targeting tumour energy metabolism for cancer treatment has been recently highlighted [3,4].

It has been widely reported that a large number of different types of cancer show abnormal metabolism of phosphatidylcholine (PC) [5,6,7,8], the most abundant phospholipid in the eukaryotic cell membrane. PC can be synthesised from choline, an organic cation that cannot freely cross the plasma membrane; so, it requires active transporters for its entry into the cell [9,10]. Upon entering the cell, choline is recruited along the Kennedy pathway, also called the CDP-choline pathway, leading to the synthesis of the phospholipid PC. An essential component of this pathway is choline kinase (ChoK), a cytosolic enzyme that catalyses the ATP-dependent phosphorylation of choline to phosphocholine (PCho) in the presence of magnesium.

In humans, the ChoK family is the result of the expression of two genes, chok-α and chok-β, which encode for three different isoforms of the enzyme: ChoKα1 (457 residues, 52 kDa), ChoKα2 (439 residues, 50 kDa) and ChoKβ (395 residues, 45 kDa). Each soluble active isoform is present as a homo- or heterodimer or as a tetramer. Hong et al. [11] provided the crystal structure of both the ChoKα1 and the ChoKβ isoform in complex with hemicholinium-3 (HC-3), a prototype for ChoK inhibitors [12]. In both structures (PDB code: 3G15 and 3LQ3), HC-3 is bound in the conserved hydrophobic groove at the C-terminal lobe of the enzyme; however, it was observed that actual inhibition occurs only when HC-3 binds to the ChoKα1 isoform and not when it binds to the ChoKβ isoform.

The ChoKα1 isoform is considered an important anticarcinogenic and antiproliferative target due to the overexpression of the chok-α gene and to the increased activity of ChoK observed in cancer cell lines [12,13,14,15]. The higher levels of ChoKα1 activity found in tumours lead to an increase in PCho and PC levels, the latter being required for the synthesis of new cell membranes during cell proliferation [13]. This high enzymatic activity, as well as the overexpression of the chok-α gene, has been closely associated with alterations in oncogenic signalling pathways, such as PI3K/AKT or MAPK/AKT [16,17]. It is worth highlighting that hypoxia-response elements are present in the promoter sequence of chok-α, where hypoxia-inducible factors (HIF-α) may bind. The binding of HIF-1α enhances the overexpression of the chok-α gene in hypoxic environments such as those occurring in tumours [18]. 

Following the identification of ChoKα1 as a bona fide target in cancer therapy, during the past decade, several research groups have synthesized compounds that are capable of inhibiting this enzyme. The first of these, HC-3, features an IC_50_ value of 500 µM for ChoK inhibition but blocks sodium-dependent choline transport and the synthesis of acetylcholine, thus showing numerous adverse effects. A symmetrical bis-pyridinium derivative, MN58b, and a symmetrical bis-quinolinium derivative RSM-932A (known as TCD-717) were subsequently synthesized, both of them showing reduced toxicity in human tumours [12]. 

Our group has long been involved in the design and synthesis of ChoKα1 inhibitors, which we have also tested for their antitumoral potency on several cancer cell lines. In the quest for more active and selective ChoKα1 inhibitors, the screening of different mono- and bis- cationic compounds showed dissociation constants (Kd) between (82–35) µM and (0.62–0.11) µM, respectively. Similar results to HC-3 (K_d_ = 0.180 ± 0.05 µM) were obtained for those biscationic molecules with longer linkers (*n* = 4) [19]. We recently studied two new symmetrical biscationic compounds—1,1′-(((ethane-1,2-diylbis(oxy))bis(4,1-phenylene)), bis(methylene))-bispyridinium or –bisquinolinium bromide, EB-3D and EB-3P, respectively, which contain a pair of oxygen atoms in the spacer between the biscationic moieties. Both compounds inhibited ChoKα1 activity and HepG2 cell proliferation at low micromolar concentrations [20]. 

To date, our group has carried out studies with ChoKα1 inhibitors that have cationic heads derived from quinolinium, isoquinolinium or pyridinium. Our results indicate that, with these heads, changes in the spacer lead to more effective inhibition of the ChoKα1 enzyme while also causing antiproliferative activity [19,21,22], albeit to variable extents in different cell lines. However, the main drawback of previously synthesised inhibitors resides in the lack of total convergence between the inhibition and antiproliferation values. Docking and SAR studies have provided evidence of the binding and activity of biscationic and symmetrical inhibitors, but biological tests have pointed out that other factors could be affecting the cell-growth outcome. 

In previous reports, we have described that some ChoKα1 inhibitors are also capable of inhibiting choline transport, which could limit the intracellular availability of choline for PC biosynthesis [20,23]. Cellular choline uptake can be carried out by active transporters, low-affinity organic cation transporters (OCTs), intermediate-affinity choline transporter-like proteins (CTLs) and high-affinity choline transporters (CHTs). CHTs transport choline by a sodium-dependent mechanism, whereas OCTs and CTLs are sodium-independent transporters [24]. CHTs are present in cholinergic neurons and participate in acetylcholine synthesis. OCTs and CTLs are present in several tissues and supply choline mainly for the synthesis of PC and other phospholipids [10]. The expression and the function of choline transporters have not been well identified in cancer, although overexpression of CTL1 has been found in malignant cells and tumours of the liver, lung, colon, breast, prostate and ovaries [12,25,26]. In addition, Watanabe et al. [27] showed that the choline transporter CTL1 is highly expressed in tumour cells and that the inhibition of CTL1 function induces apoptotic cell death, making CTL1 a potential target in cancer therapy.

Our goal is to obtain more potent and selective ChoKα1 inhibitors that also feature increased antiproliferative activity in cell lines of different origins. For this reason, we focus on the synthesis of symmetrical bioisosteric molecules with different electron donor or acceptor groups in the linker, with the purpose of (1) increasing the binding interaction with the enzyme and (2) improving key parameters such as the solubility of the inhibitors. In this study, a new ChoKα1 inhibitor called PL48 (Figure 1) is presented. Owing to the presence of sulphur atoms in the linker, PL48 shows a higher lipophilicity than its predecessor EB-3P (Log *p* = 7.07 vs. 6.03, both values calculated using http://www.swissadme.ch/ (last access the 29 December 2021) as an average of five prediction methods). Moreover, sulphur free-electron pairs allow the establishment of new interactions within the choline-binding pocket of ChoKα1, improving its affinity for the enzyme. Preliminary docking studies predicted the dithioethane linker to sit in a more polar interloop region (L1-L9) with which it can interact. In comparison, the crystal structure of the diphenoxyethane homologous crystal structure (PDB code: 5FTG) showed the allocation of the linker in a more hydrophobic cage, mostly interacting through the phenyl group via π-π stacking with Tyr354 and Phe435. In addition, the sulphur-containing linker, even when most exposed, leads to less solvation than do the most hydrophilic oxygen and thus to better interactions with the enzyme.

No PAINS (Pan Assay Interference Structures) were detected for the new inhibitor, ex-cluding the possibility of side effects due to unspecific activity of the chemical structure.

In the present study, we examined the effect of PL48 on ChoKα1 expression and activity, as well as on the activity of the main choline transporters involved in the choline uptake in HepG2 and MCF7 cancer cell lines. Our objective was to determine the correlation between ChoKα1 activity and choline uptake on cell proliferation.

## 2. Materials and Methods

### 2.1. Materials

Foetal Bovine Serum (FBS), Eagle’s Minimum Essential Medium (MEM) and RPMI-1640 (Roswell Park Memorial Institute 1640) were obtained from Biowest (Nuaillé, France). Thin Layer Chromatography (TLC) plates and protease inhibitor cocktail were from Sigma-Aldrich (Madrid, Spain). [Methyl-^14^C]choline was from Perkin Elmer (Madrid, Spain). Mini-PROTEAN^®^ TGX Stain-Free Protein Gels, Trans-Blot Turbo Mini PVDF and Clarity Western ECL substrate were from Bio-Rad Laboratories, Inc. (Madrid, Spain). Monoclonal anti-human primary antibodies ChoKα (sc-23382) and polyclonal β-actin were from Santa Cruz Biotechnology, Inc. (Heidelberg, Germany). Rabbit polyclonal SLC44A1/CTL1 antibody (ab110767) was from Abcam (Cambridge, MA, USA). Horseradish peroxidase (HRP)-linked secondary IgGs were from Cell Signaling Technology (Danvers, MA, USA). The pET-28a vector and *Escherichia coli* BL21 (DE3) Star cells were from Invitrogen (Carlsbad, CA, USA), N-terminal 6x His-tag was purchased from Genescript (Piscataway, NJ, USA), Ni-NTA agarose beads were from Qiagen (Venlo, The Netherlands), and HiPrep 26/60 Sephacryl 100 HR column was from GE Healthcare (Little Chalfont, Buckinghamshire, UK). All other reagents were of analytical grade.

### 2.2. Cell Culture

The breast cancer cell line MCF7 (Michigan Cancer Foundation-7) and the liver cancer cell line HepG2 (Hepatoblastoma G2) were provided by the European Collection of Animal Cell Cultures (Salisbury, UK). MEM and RPMI-1640 were used to culture HepG2 and MCF7, respectively, and were both supplemented with 10% (*v*/*v*) heat-inactivated FBS, penicillin (100 IU/mL) and streptomycin (100 μg/mL). L-glutamine at a concentration of 2 mM was also exogenously added. Cells were incubated at 37 °C in a humidified atmosphere with 5% CO_2_. When the cells reached high confluence, subculturing was carried out in a fresh medium.

### 2.3. Choline Uptake Assays

To study choline uptake, 300,000 MCF7 or HepG2 cells/well were seeded in 12-well plates. After 24 h, the culture medium was carefully aspirated, and the wells were washed twice with sodium-free buffer (SFB). This buffer contained 280 mM D-mannitol, 4.8 mM KCl, 1.2 mM CaCl_2_, 1.2 mM KH_2_PO_4_, 5.6 mM glucose, 1.2 mM MgSO_4_ and 25 mM HEPES (pH 7.4). Then, 250 µL of SFB containing isotopically labelled choline was added to each well (36 µM, 55 Ci/mol). 

For kinetic studies, when it was necessary, unlabelled choline was added in different amounts to obtain the required concentrations. The range of choline concentrations used for the assays was 1 to 1000 μM. The plates were incubated for 10 min at 37 °C and then placed on ice to block choline uptake by cells. After this, the medium was removed, and the wells were washed twice with concentrated unlabelled choline (580 μM). Subsequently, the cell monolayer was solubilised in NaOH 0.1 N, and aliquots were taken to measure radioactivity in a Beckman liquid scintillation counter (Model LS-6000-TA, Beckman, Madrid, Spain). From the analysis of the results, Michaelis–Menten ([choline] vs. v) and Eadie–Hofstee graphs (v/[choline] vs. v) were obtained to determine the maximum velocity (V_max_) and the Michaelis constant (K_M_) for the choline transport kinetics. 

In parallel, to better understand the choline transporters involved in choline uptake, we measured the incorporation of choline into cells in the presence of increasing concentrations of well-known choline transporter inhibitors, HC-3 and tetraethylammonium (TEA), or in the presence of PL48 synthesised as a ChoKα1 inhibitor. HC-3 and TEA are validated inhibitors of the CHT/CTL and OCT proteins, respectively. Samples were processed as described above. 

### 2.4. Cloning, Protein Expression and Purification of ChoKα1

A truncated form of ChoKα1 (Δ75–457) cloned into a pET-28a vector and featuring an N-terminal 6x His-tag was used to transform *E. coli* BL21 (DE3) Star cells. The transformed cells were cultured in Luria–Bertani (LB) medium at 37 °C until OD_600_ = 0.6. After induction with 1 mM isopropyl β-D-1-thiogalactopyranoside (IPTG), the bacterial cell culture was grown overnight at 20 °C and 180 rpm. The cellular pellet was then separated from the exhausted medium by centrifugation at 10,000 rpm, resuspended in 50 mM Tris-HCl pH 7.5, 500 mM NaCl, 0.2 mM phenylmethylsulphonyl fluoride (PMSF), DNase and 0.5 mM β-mercaptoethanol, and sonicated. The soluble fraction containing the enzyme was separated from the insoluble fraction by centrifugation at 15,000 rpm and 4 °C. A two-step purification protocol was employed to isolate the target enzyme. The first step was performed using Ni-NTA affinity chromatography. The cell lysate was first incubated for 45 min with Ni-NTA agarose beads. Then, the column was extensively washed with 40 column volumes (CV) of 50 mM Tris-HCl pH 7.5, 300 mM NaCl, 10 mM imidazole and 1 CV of 50 mM Tris-HCl pH 7.5, 300 mM NaCl and 40 mM imidazole. Finally, the His-tagged enzyme was eluted from the column with 50 mM Tris-HCl pH 7.5, 300 mM NaCl and 400 mM imidazole. The second purification step was performed by size-exclusion chromatography using a HiPrep 26/60 Sephacryl 100 HR column (GE Healthcare, Little Chalfont, Buckinghamshire, UK), which was previously equilibrated with 20 mM Tris/HCl pH 7.5 and 150 mM NaCl running buffer. After the two purification steps, a highly pure sample was obtained, and the final enzyme yield was 1.25 mg of recombinant protein per litre of bacterial culture.

### 2.5. Preparation of Cell Lysate Containing Soluble ChoKα

MCF7 or HepG2 cells were seeded at a concentration of 1 × 10^6^ cells/well in 6-well plates. After seeding, the medium was removed and replaced with a fresh medium containing 1 μM PL48 or none as a control. After 48 h, cell lysate was obtained as described by Jiménez-López et al. [28] with minor modifications. Briefly, cells were scraped into PBS and centrifuged at 1500 rpm for 5 min. The cell pellet was resuspended in 100 mM Tris-HCl pH 8.5 and sonicated for 3 s with a microprobe in an ice bath, and the lysate was immediately frozen in liquid nitrogen and stored at −80 °C until use. 

### 2.6. Determination of ChoKα Activity

To study the effect of PL48 on ChoKα, an in vitro enzymatic assay was performed with the purified enzyme or the cell lysate containing the enzyme, as previously reported by Schiaffino-Ortega et al. [21] and by Schiaffino-Ortega et al. [29], respectively. Briefly, the incorporation of ^14^C from [methyl-^14^C]choline into PCho in either the absence (control) or presence of different PL48 inhibitor concentrations was used to determine the ChoK activity. For the purified enzyme assay, the reaction mixture included 20 ng of purified ChoKα1, 10 mM ATP, 10 mM MgCl_2_, 100 mM Tris-HCl pH 8.5 and increasing concentrations of PL48. Samples were preincubated at 37 °C for 5 min, and then a pulse of [methyl-^14^C]choline chloride (1 mM, 4500 dpm/nmol) was added, and the reaction proceeded for 10 min at 37 °C. When determining ChoKα activity in cell lysate, the procedure was similar, but 50 μg of protein was added, and after preincubation at 37 °C for 5 min, the reaction was initiated with 1 mM [methyl-^14^C]choline (4500 dpm/nmol) and was incubated at 37 °C for 20 min.

The assay was stopped by immersing the reaction tubes in boiling water for 3 min. Aliquots of the supernatant were applied to the origin of silica gel plates in the presence of PCho (0.1 mg) and choline (0.1 mg) as carriers. The TLC plates were developed in methanol/0.6% NaCl/28% NH_4_OH in water (50:50:5, *v*/*v*/*v*) as solvent. PCho was visualised under exposure to iodine vapour, and the corresponding spot was scraped and transferred to scintillation vials for measurement of radioactivity by a Beckman 6000-TA (Madrid, Spain) liquid scintillation counter. The 50% inhibitory concentrations (IC_50_ values) were determined from the % enzyme activity at different concentrations of synthetic inhibitors relative to the control by using a sigmoidal dose–response curve (ED50plus v1.0 software).

### 2.7. Evaluation of the Antiproliferative Effect of PL48 in HepG2 and MCF7 Cell Lines

HepG2 and MCF7 were seeded onto 96-well plates (10 000 cells/well) and maintained in medium for 24 h. Then, the culture medium was replaced with fresh medium, and the cells were incubated for 24 or 48 h in the absence or presence of different amounts of PL48. The antiproliferative effect of PL48 was evaluated by crystal violet staining assay using a cell-number-based standard curve, as previously reported [30]. The absorbance of crystal violet in each well was measured at a wavelength of 590 nm directly in plates using a Synergy ^™^ HTX Multi-Mode microplate reader through Gen5^®^ software (BioTek, Oxford, UK).

### 2.8. Immunoblotting Assay

MCF7 or HepG2 cells growing in the log phase were incubated for 48 h with medium in the absence or presence of 1 μM PL48. Then, the cell monolayers were washed with cold PBS and subsequently scraped in PBS, followed by centrifugation (2 500 rpm/5 min/4 °C). The pellet was resuspended in lysis buffer, which contained 50 mM Tris-HCl, pH = 7.4, 150 mM NaCl, 1% Triton X-100, protease inhibitor cocktail, 1 mM sodium orthovanadate and 57.4 mM PMSF. The tubes were kept on ice for 30 min accompanied by vortexing every 5 min. Then, a second centrifugation was performed (13 000 rpm/15 min/4 °C), and the supernatants were collected and stored at −80 °C. Equal protein amounts from lysates were separated by SDS-PAGE and transferred to PVDF membranes. Prestained protein molecular weight markers were used. Membranes were blocked in 5% non-fat dried milk in TBS and 0.05% Tween-20 in TBS for 1 h and incubated with CTL1 (1:500) or ChoK (1:200) primary antibodies. The corresponding horseradish peroxidase (HRP)-conjugated IgG (1:5000) was used as a secondary antibody and incubated for 1 h. Immunoreactive proteins were detected using ECL substrate, and the membranes were imaged using the Molecular Imager ChemiDocTM MP System (Bio-Rad Laboratories, Inc., Madrid, Spain).

### 2.9. Metabolic Labelling Assays

MCF7 and HepG2 (300,000 cells/well) were incubated in both the presence and absence of 1 μM PL48 for 48 h. [Methyl-^14^C]choline (60 μM, 33 Ci/mol) was added in the last 4 h of the incubation period. Lipid biosynthetic activity was estimated according to the level of incorporation of radiolabelled choline into corresponding metabolites of PC synthesis. Water-soluble and lipid fractions were extracted from the cells following the procedure of Bligh and Dyer [31]. PC and sphingomyelin (SM) were separated from the chloroformic phase on silica-gel 60 G TLC plates using a mixture of chloroform/methanol/acetic acid/water (60:30:8:5, *v*/*v*) as a solvent. Choline and PCho were separated from the water–methanol phase by TLC using a solvent of methanol/0.6% NaCl/NH_4_ (10:10:1, *v*/*v*). The spots were made visible by exposure to iodine vapour and radiometrically measured by liquid scintillation using a Beckman 6000-TA counter (Madrid, Spain).

### 2.10. Other Analyses

Cell protein content was determined using the method of Bradford [32] with BSA as the standard.

### 2.11. Statistical Analysis

The results are expressed as means ± SEM (standard error of the mean). A one-way ANOVA was conducted with post hoc comparisons by Scheffé’s test (SPSS 13.0) using GraphPad Software Inc. (San Diego, CA, USA). *p*-value < 0.05 is considered statistically significant. Asterisks indicate the following *p*-value ranges: * *p* < 0.05, ** *p* < 0.001, *** *p* < 0.0001.

## 3. Results and Discussion

### 3.1. Functional Characteristics of Choline Uptake in MCF7 and HepG2 Cells

Choline is an organic cation constituent of choline-containing lipids such as PC and SM, lipoproteins, bile lipids and pulmonary surfactant [33]. Cell membranes are not permeable to choline [9,34]; thus, in this work, we characterised the kinetics of choline uptake by MCF7 and HepG2 cancer cell lines. For this purpose, cells were incubated for 10 min with [methyl-^14^C]choline with substrate concentrations ranging between 1 and 1000 µM in the absence of sodium. In both cell lines, plots of the uptake kinetics exhibited a typical hyperbolic appearance (Figure 2A). The Eadie–Hofstee plots showed a concave appearance with two straight lines with different slopes, indicating clearly that, in both cell types, two different choline transport systems were involved (Figure 2B,C, left graphs). The kinetic constants determined using the initial rates obtained with a low concentration of substrate (1–36 μM) (middle plots) were: K_M_ = 5.38 μM and V_max_ = 0.85 nmol/min/mg protein for HepG2 cells. Similar values were obtained in MCF7 cells (K_M_ = 5.67 μM and V_max_ = 0.37 nmol/min/mg protein). When using higher choline concentrations (60–1000 μM) (right graphs), the kinetic constants for HepG2 cells (K_M_ = 183.57 μM and V_max_ = 2.75 nmol/min/mg protein) and MCF7 cells (K_M_ = 137.16 and V_max_ = 2.48 nmol/min/mg protein) were also similar. These results reveal that in both HepG2 and MCF7 cancer cell lines, K_M_ values for transporter-mediated uptake were in the intermediate-affinity and low-affinity ranges [10]. Previously reported data in several cell lines suggested that two different transport systems with different affinities for choline cooperate to enable choline entry into the cell [35]. In several tissues, the involvement of low-affinity and intermediate-affinity choline transporters has also been described [35,36].

### 3.2. Effect of HC-3 and TEA on Choline Uptake in MCF7 and HepG2 Cells

HC-3 and TEA are validated inhibitors of the CHT/CTL and OCT proteins, respectively. Therefore, to further characterise the choline transport systems, we carried out competition assays with these inhibitors in both cell lines. Measures of inhibition using HC-3, a well-known competitive inhibitor of high-affinity choline transport at low concentrations (<1 μM), indicated that choline uptake was not inhibited. However, treatments with higher concentrations of HC-3 to target other HC-3-sensitive transporters, such as CTL1-mediated transport, caused the blockage of choline uptake compared with control cells. Inhibition using HC-3 concentrations ranging from 2 μM to 150 μM resulted in IC_50_ values of 29.42 ± 0.97 μM for HepG2 cells and 39.06 ± 6.95 μM for MCF7 cells (Figure 3A and 3B, respectively). These results exclude the participation of the sodium-dependent CHT1 transporter, which is expressed mainly in neuronal tissue, very sensitive to HC-3, and a limiting step in acetylcholine biosynthesis in cholinergic neurons. In the cells used in our study, experimental data indicate that the transporter involved could be CTL1, which has an intermediate affinity for choline, with a K_M_ in the low micromolar range. Similar to the high-affinity choline transporter CHT1, CTL1 is selectively inhibited by HC-3 but with a lower level of sensitivity (10–100 µM) [9].

The inhibitor of the OCT family of transporters, TEA, showed significant effects on choline uptake only at very high concentrations. As shown in Figure 3, TEA only caused weak inhibition, showing IC_50_ values at a concentration greater than 1 mM; hence, OCT-mediated choline uptake in HepG2 and MCF7 must be negligible (Figure 3C,D), and it is probable that OCTs do not significantly contribute to choline uptake in these cell types. These data agree with results previously reported by Sinclair et al. [36] in the liver and those reported by Morse et al. [37], who described the low expression of OCT transporters in MCF7 cells. In summary, this study suggests the functional role of CTL1 as a choline transporter in MCF7 and HepG2 cells. Thus, we focused our study on the CTL1 choline transporter, which is expressed in different organisms and cell types, apparently not for the biosynthesis of acetylcholine but for the production of the most abundant metabolite of choline, the membrane lipid PC [38,39]. 

### 3.3. PL48 Inhibits Cell Growth in MCF7 and HepG2 Cells

In tumour cells and in tumour progression, PC biosynthesis is greater than in normal tissue. Furthermore, the overexpression of the ChoKα1 isoform and CTL1 has been found in several malignant cells and tumours [12,25,26]. This suggests that the metabolism of choline and related compounds is a metabolic hallmark of tumour onset and progression. In light of such data, we previously synthesised ChoKα1 inhibitors that showed antiproliferative activity in different cell lines [23,40]. One series of such inhibitors comprised symmetrical biscationic compounds, bis-pyridinium and bis-quinolinium derivatives with 1,2-diphenoxyethane as a spacer between the bi-pyridine or bi-quinoline rings. Among this last series of compounds, it is worth highlighting compounds 10a (also called EB-3D) and 10l (called EB-3P), which inhibit ChoKα1 with similar IC_50_ values of around 1 μM and show GI_50_ values in the HepG2 cell line of 14.55 and 4.81 μM, respectively [20].

PL48 is a bioisostere of the ChoKα1 inhibitor EB-3P. As can be seen from its structure (Figure 1), PL48 contains a pair of sulphur atoms in the spacer of the linker between the biscationic moieties, while EB-3P has two O-atoms [20]. The presence of sulphur atoms between the quinoline cationic heads increases its lipophilicity. We evaluated the dose-dependent antiproliferative activity of PL48 against the HepG2 and MCF7 cell lines, identifying GI_50_ values of 0.972 ± 0.240 μM for HepG2 and 1.34 ± 0.480 μM for MCF7 after 48 h of treatment (Figure 4A and 4B, respectively). This antiproliferative action could not be attributed to lysis as measured by LDH release into the medium (data not shown). Our results indicate that in these cell lines, (i) PL48 had higher antiproliferative activity than its bisquinolinium predecessor EB-3P, which is a ChoKα1 inhibitor previously studied by our group [20], and (ii) both cell lines show the same sensitivity to PL48.

### 3.4. Inhibition of ChoKα1 by PL48

In order to assess the efficacy of PL48 in inhibiting the target protein for which it has been designed, the effect of PL48 on human recombinant ChoKα1 was assayed by determining the rate of incorporation of ^14^C from [methyl-^14^C]choline into PCho in the absence or presence of different PL48 concentrations, as previously described [21,40]. As shown in Figure 5, PL48 inhibits ChoKα1 activity with an IC_50_ value of 0.658 ± 0.067 μM, below that of previously studied symmetrical biscationic compounds [21]. 

In Figure 6, the improved potency of the inhibitor that contains the dithioethane linker is shown. Sulphur bioisosteric substitution confers a more lipophilic character to the molecule.

### 3.5. Effect of PL48 on Choline Uptake in HepG2 and MCF7 Cell Lines

As mentioned above, we have described that some ChoKα1 inhibitors are also capable of inhibiting choline transport [20,23]. According to our results, in cells treated for 10 min with the inhibitor PL48, choline uptake was strongly inhibited, featuring IC_50_ values of 0.09 ± 0.03 μM for HepG2 and 0.26 ± 0.01 μM for MCF7 (Figure 7). 

### 3.6. Effect of PL48 Treatment for 48 h on ChoKα and CTL1 Expression Levels in MCF7 and HepG2 Cells

Previous studies have reported the overexpression of ChoKα in several cancer-derived cell lines [43,44]. ChoK inhibition or ChoKα downregulation leads to a decrease in PCho levels in the malignant cells, coupled with a decrease in cell proliferation [5,15,45,46,47], demonstrating a causal link between ChoKα elevation and carcinogenesis and tumour progression. 

In this context, our research group showed that EB-3D and EB-3P act as ChoKα1 inhibitors and decrease the expression levels of this enzyme [20]. Some works, however, have shown that the inhibition of the enzyme in MCF7 cells produces an increase in the expression of both CTL1 and ChoK, raising PCho levels, which is used as a noninvasive response biomarker after cancer treatment [48]. These results are controversial, since elevated PCho levels are generally associated with cancer progression [5,15,45,46,47]. 

In the present study, we analyse the expression levels of ChoKα and CTL1 in both HepG2 and MFC7 cells after treatment with PL48 for 48 h. To assess the effect of PL48 on the levels of these two proteins in both cell types, we treated cells with PL48, and cell lysates were prepared as described in the Materials and Methods section. The images of ChoKα expression levels from HepG2 cells and MCF7 are represented in Figure 8. As shown in the figure, in control cells, ChoKα levels are higher in MCF7 than in HepG2. The literature includes works that support these results, such as those presented by See Too et al. [49], who detected α1 and α2 isoforms of ChoK in several cancer cell lines. These authors observed that MCF7 cells showed higher expression of total ChoKα compared with HepG2 cells [49]. However, when analysing the expression profiles of ChoKα and ethanolamine kinase, Ling et al. [50] observed that ChoKα was expressed at lower levels in MCF7 cells compared with the HepG2 cancer cell line. In any case, we show that treatment of HepG2 and MCF7 cells with PL48 resulted in a 43% or 22% reduction in ChoKα protein levels, respectively (Figure 8).

Furthermore, as indicated in Figure 8, CTL1 was highly expressed in HepG2, and the levels of CTL1 were also significantly lower after HepG2 or MCF7 cells were exposed to PL48 for 48 h. We can conclude that the use of ChoKα and CTL1 as therapeutic targets in these cancer cells is an appropriate approach to decrease cell proliferation due to the relationship between cell growth and choline metabolism. 

### 3.7. Effect of PL48 Treatment for 48 h on ChoKα and CTL1 Activities in MCF7 and HepG2 Cells

In order to confirm the decrease in the expression and activity of the CTL1 transporter after treatment with PL48, we conducted a metabolic assay. Cells were incubated with the inhibitor for 48 h, and after its withdrawal, were incubated for 10 min with [methyl-^14^C]choline and choline uptake was determined. The results obtained show that in both types of cells, treatment with PL48 for 48 h inhibited the uptake of choline (Figure 9A). This inhibition is not produced by the direct action of the inhibitor on the transporter since PL48 was removed before the uptake assay; instead, inhibition of choline uptake appears to be the result of transporter activity after treatment with PL48 and can be considered an indirect measure of its expression. These data agree with the results obtained from the Western blot, which indicated a significant decrease in the expression levels of the CTL1 transporter.

Furthermore, experiments were performed to assess the total cellular activity of ChoK as previously described [29]. In brief, we obtained cell lysate containing the ChoK enzyme after treating cells with the inhibitor for 48 h, observing a decrease in the activity of the enzyme in cells incubated with PL48 (Figure 9B). The results again confirm that PL48 results in the lower expression of ChoK.

### 3.8. Effect of PL48 on Choline Metabolism in MCF7 and HepG2 Cells

After establishing that PL48 caused the marked inhibition of recombinant human ChoKα activity and strongly inhibited [^14^C]choline uptake in MCF7 and HepG2 cells in vitro, we explored the effects of the long-term exposure of these cells to PL48 on choline metabolism. To this end, we exposed the cells to this compound for a 48 h period and analysed the distribution of total radiolabelled choline incorporated into the cell.

In these experiments, choline-labelled metabolites were extracted from the cells with a mixture of aqueous and organic solvents to determine the distribution of choline incorporated into the two phospholipids, PC and SM, and water-soluble metabolites. As can be seen in Table 1, HepG2 cells incorporated significantly higher levels of radioactivity than MCF7 cells, which is in accordance with the results of Western blot experiments, showing that choline uptake is more active in the hepatoma cell line. It is noteworthy that in this cell line, approximately 77% of the intracellular radioactivity was associated with the aqueous phase, whilst the lipid fraction accounted for only 23% of the total radioactivity incorporated into the cells. In MCF7 cells, 16% of the labelling was observed in the lipid-soluble fraction.

We also determined the radioactivity associated with both choline and PCho in the water-soluble phase. It is remarkable that in both cells, PCho was the main metabolite (85–95% of total label) found in the aqueous phase, whilst choline did not account for more than 5–15% of this fraction, indicating a high rate of phosphorylation of [^14^C]choline incorporated by the cell (data not shown). In the organic phase of lipids containing choline, we observed that PC accounted for more than 98% of the total radioactivity in the organic phase in both cell lines (data not shown). These results agree with those obtained by other authors who also showed that only a minor fraction of radiolabelled choline incorporated by different cell types is present as phospholipids [51,52].

Our results show that choline metabolism responds differently in HepG2 and MCF7 after prolonged exposure to PL48 (Table 1). In particular, 48 h of PL48 treatment significantly reduced the total incorporation of [^14^C] choline into cells, but the effect was more marked in HepG2 (35% reduction in total choline incorporated by the cell) in comparison with MCF7 cells (25% reduction). This reduction, besides the downregulation of CTL1 mentioned above, clearly suggests that the apparent choline uptake is significantly diminished in cells chronically treated with the inhibitor.

Due to the decrease in choline uptake, we observed that the radioactivity associated with PCho after chronic PL48 treatment in MCF7 cells was significantly reduced (up to 30%) with respect to that observed in control cells. Similarly, according to the drastic reduction in choline transport produced by PL48 treatment, the radiolabel of PCho dropped by up to 31% compared with control HepG2 cells. In addition, it is remarkable that the exposure of HepG2 to PL48 led to a drastic reduction (up to 53%) in PC formation, whilst the PC radiolabel was unchanged in MCF7 when compared with control cells. These results agree with the suggestion that in HepG2 cells, inhibition of both ChoK and choline uptake contributes to a drop in PC levels, whilst in MCF7 the decrease in choline uptake is not reflected in a reduction in PC formation, presumably because since ChoK is only slightly affected by chronic PL48 treatment.

## 4. Conclusions

The results reported herein clearly demonstrate that PL48, a rationally designed ChoKα1 inhibitor, inhibits the activity of human recombinant ChoKα1 as well as choline uptake after 10 min exposure in both HepG2 and MCF7 cells in vitro. The results describe the kinetics of choline uptake and identify the choline transporter CTL1 as the main carrier responsible for supplying choline for PC synthesis in these two types of cells. However, treatment with PL48 for 48 h exerts a differential effect on the two cell lines. In HepG2, we observed the inhibition of choline uptake and CTL1 expression as well as inhibition of the activity and expression of ChoK. Therefore, it is likely that both mechanisms, namely, targeting choline uptake and ChoK activity, are responsible for the inhibition of PC synthesis and, consequently, that the efficacy of this inhibitor on cell proliferation is closely correlated with its capability to block choline uptake and ChoKα activity. In MCF7 cells, the results lead us to conclude that the decrease in choline intake after treatment with PL48 for 48 h is mainly responsible for the antiproliferative activity produced by the inhibitor in these cells. These cells express ChoK, but the decrease in its activity and expression after treatment with PL48 for 48 h is less than that observed in HepG2.

The results obtained in this work indicate that PL48 inhibits not only Chokα1 activity and expression but also the uptake of choline through the CTL1 transporter (Figure 10). Both ChoKα1 and CTL1 proteins are promising therapeutic targets to inhibit cell proliferation in MCF7 and HepG2 cell lines. Our data show that PL48 is a good candidate for further preclinical evaluation as a potential anticancer drug.

## Figures and Tables

**Figure 1 pharmaceutics-14-00426-f001:**
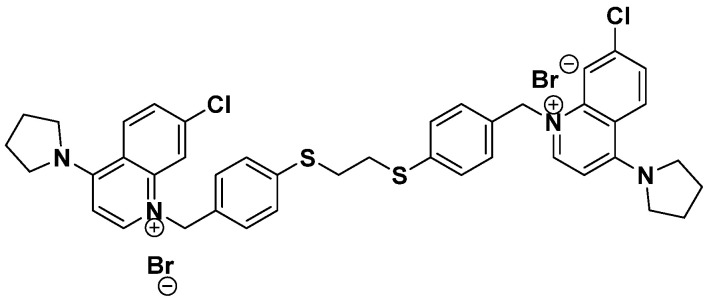
Chemical structure of synthetic ChoKα1 inhibitor PL48.

**Figure 2 pharmaceutics-14-00426-f002:**
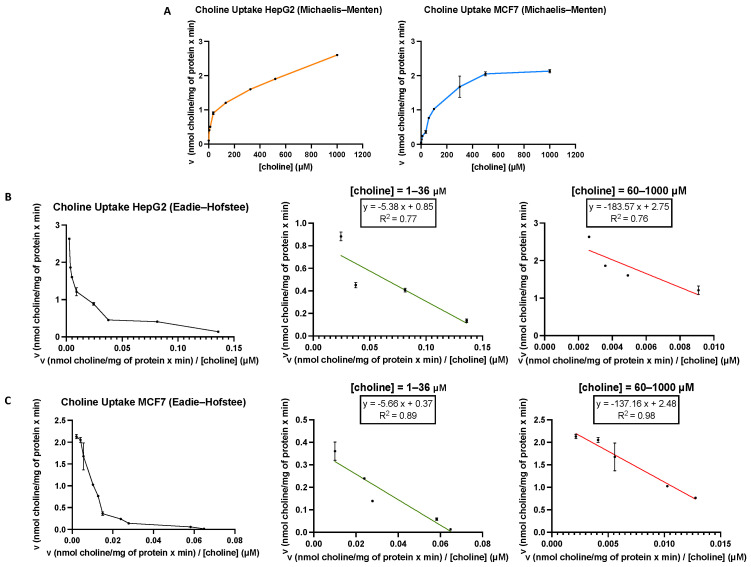
(**A**) [Methyl-^14^C]choline uptake by HepG2 and MCF7 cells was assayed for 10 min over a concentration range from 1 to 1000 μM choline. Representative Eadie–Hofstee plots for [methyl-^14^C]choline uptake in HepG2 (**B**) and MCF7 (**C**). The kinetic constants (K_M_ and V_max_) for transporter proteins were determined using equations from the Eadie–Hofstee plots obtained with low (1–36 μM) and high (60–1000 μM) choline substrate concentrations. Data are representative of two independent experiments performed in triplicate.

**Figure 3 pharmaceutics-14-00426-f003:**
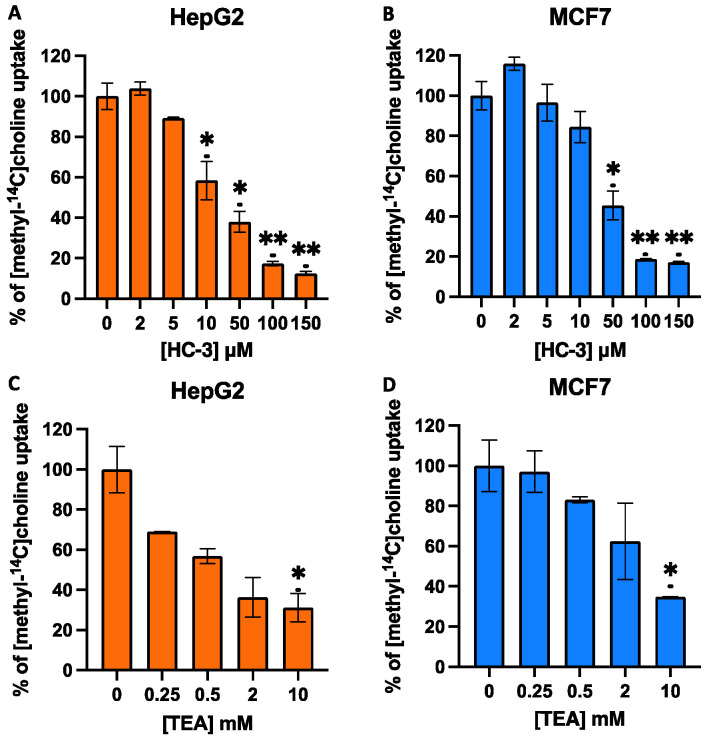
Effects of hemicholinium-3 (HC-3) (**A**,**B**) or tetraethylammonium (TEA) (**C**,**D**) on choline uptake in HepG2 and MCF7 cell lines. Choline uptake was assayed in cells treated for 10 min with increasing concentrations of inhibitors dissolved in sodium-free buffer (SFB). Data represent the mean ± SEM of two independent experiments conducted in triplicate. Results are normalised to their respective controls. * *p* < 0.05, ** *p* < 0.001.

**Figure 4 pharmaceutics-14-00426-f004:**
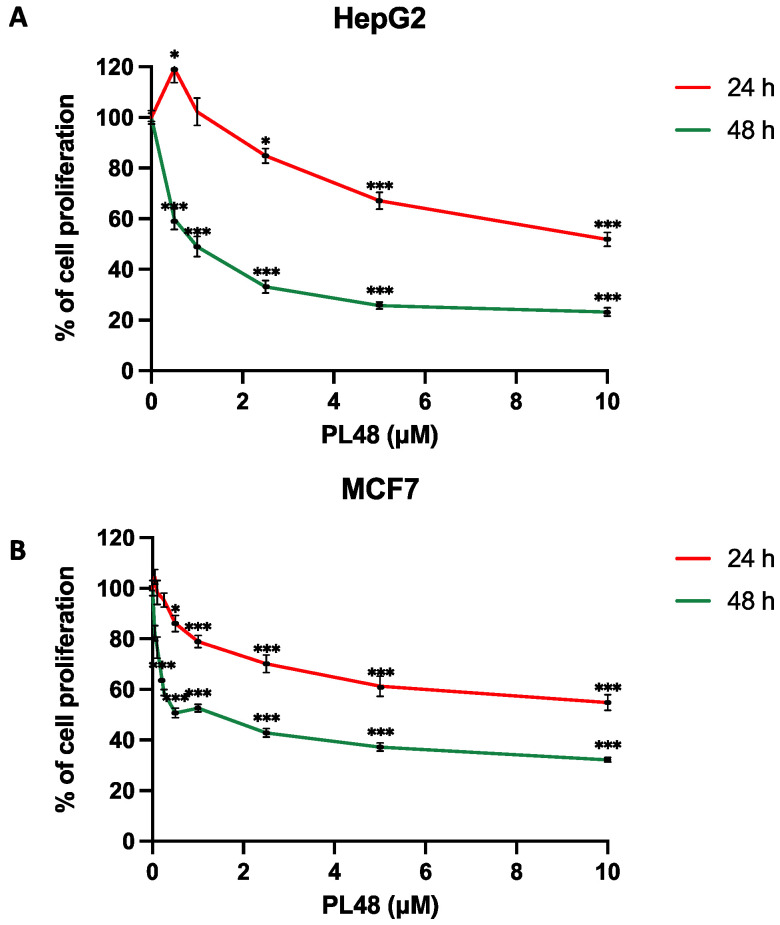
Effects of PL48 inhibitor on (**A**) HepG2 and (**B**) MCF7 cell proliferation. Cells growing in the log phase were incubated with MEM (HepG2) or RPMI-1640 (MCF7) in the presence or absence of PL48 at concentrations of up to 10 μM for 24 or 48 h. Cell number was determined by crystal violet staining and expressed as a percentage of the control cells. These experiments were performed twice in triplicate. * *p* < 0.05, *** *p* < 0.0001.

**Figure 5 pharmaceutics-14-00426-f005:**
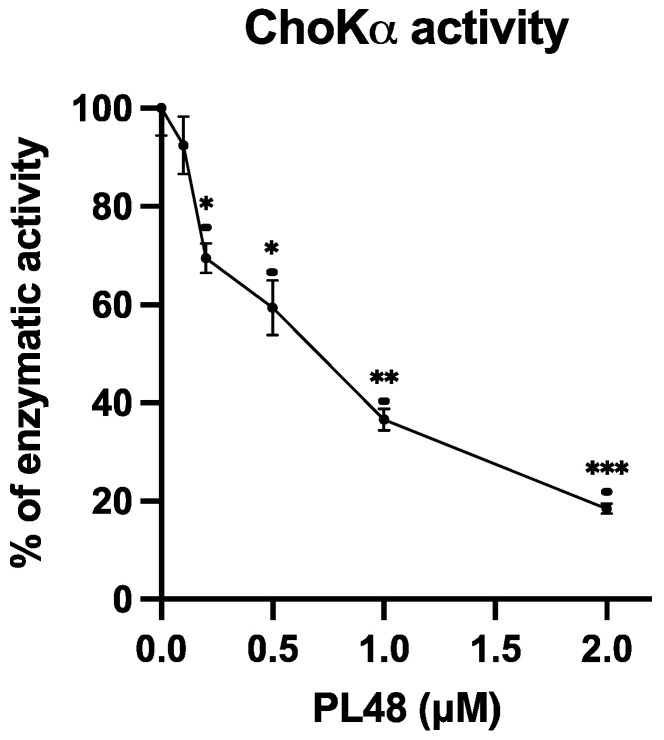
Effects of PL48 on choline kinase (ChoK)α1 activity. ChoKα1 activity was determined by the incorporation rate of ^14^C from [methyl-^14^C]choline into phosphocholine (PCho) in the absence or presence of different PL48 concentrations. The results are expressed as the percentage of enzymatic activity compared with the control. * *p* < 0.05, ** *p* < 0.001, *** *p* < 0.0001.

**Figure 6 pharmaceutics-14-00426-f006:**
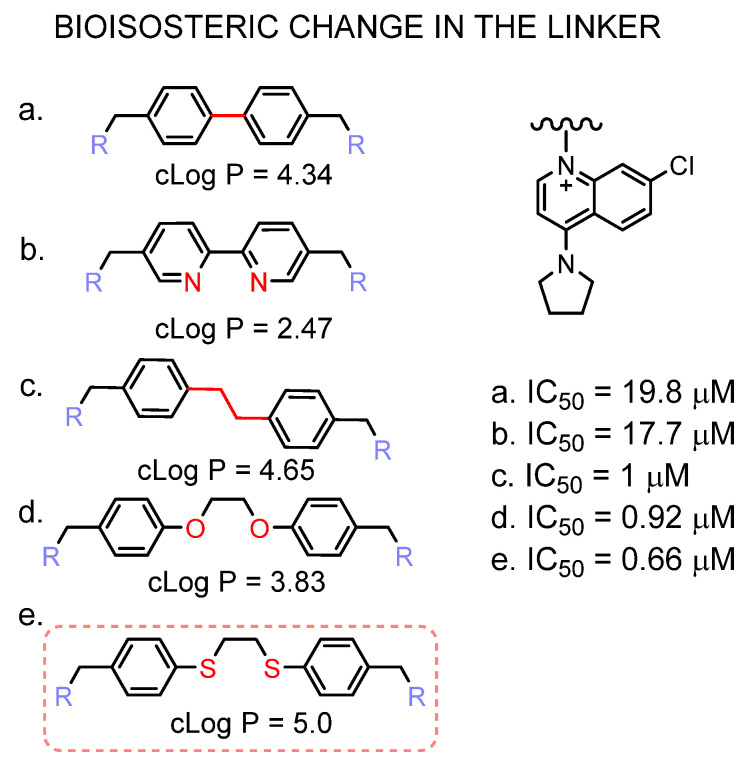
Effect of the linker bioisosteric change in the inhibitory activity towards the ChoKα1. The IC_50_ values of compounds (**a**–**d**) are taken from [21,41,42], (**e**) corresponds to PL48.

**Figure 7 pharmaceutics-14-00426-f007:**
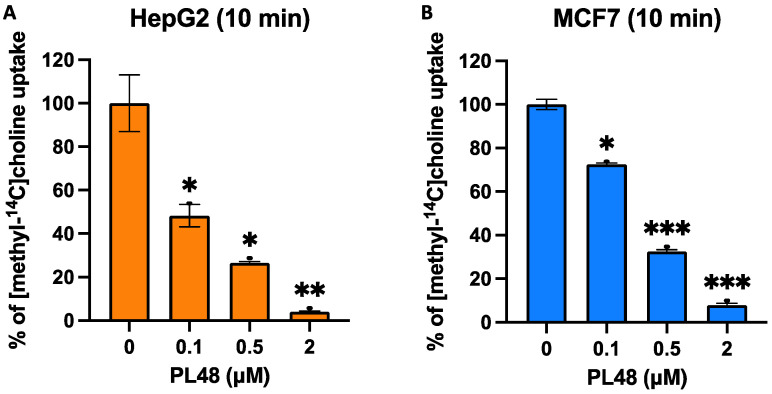
Effects of PL48 on choline uptake in HepG2 (**A**) and MCF7 (**B**) cancer cell lines. Choline uptake was assayed in cells treated for 10 min with increasing concentrations of inhibitor. Data represent the mean ± SEM of two independent experiments conducted in triplicate. Results are normalised to their respective controls. * *p* < 0.05, ** *p* < 0.001, *** *p* < 0.0001.

**Figure 8 pharmaceutics-14-00426-f008:**
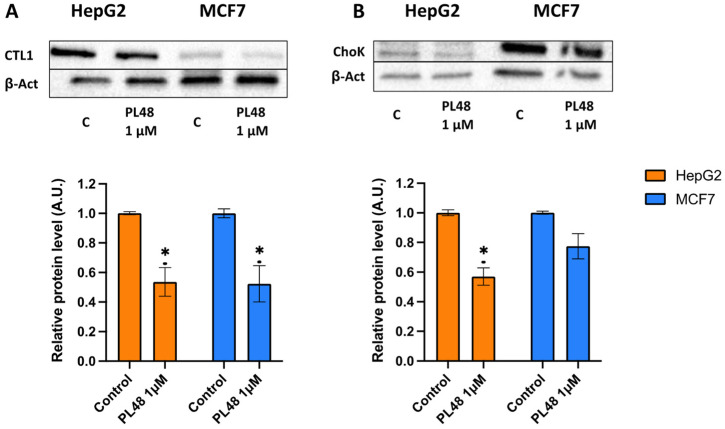
Effect of PL48 on CTL1 and ChoK protein levels in HepG2 and MCF7 cells. Cells were incubated without PL48 (control) or with 1 μM PL48 for 48 h. (**A**) Western blots show representative results of experiments repeated three times. (**B**) Protein levels in the samples were normalised to their respective β-actin levels and expressed as x-fold change compared with the corresponding control ratio (1.0). Data represent the mean ± SEM of three independent experiments. * *p* < 0.05.

**Figure 9 pharmaceutics-14-00426-f009:**
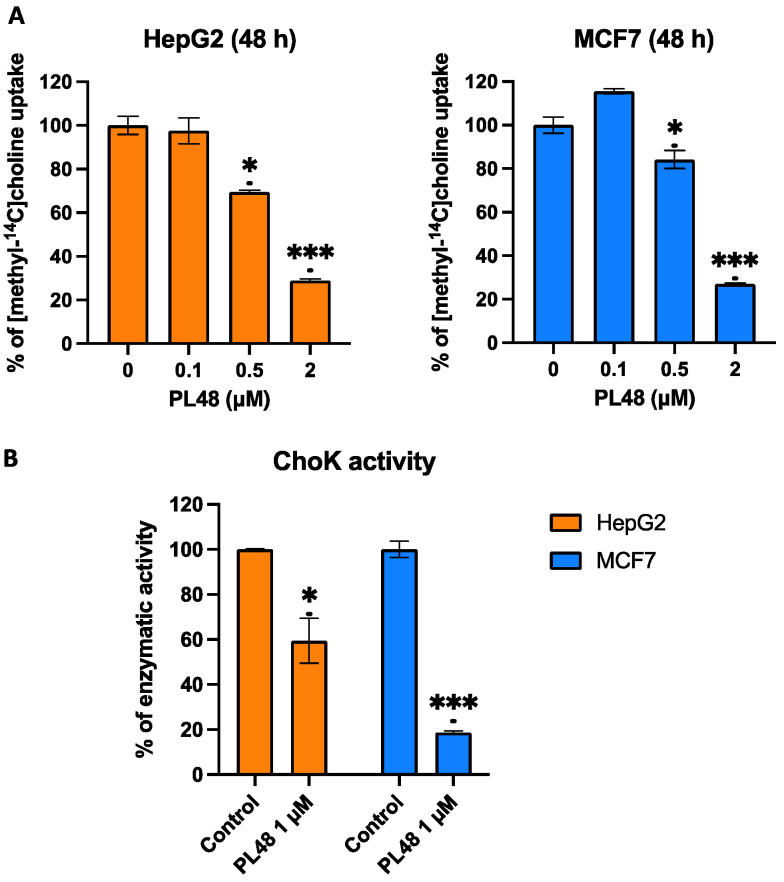
Effects of PL48 treatment on choline uptake (**A**) and on total cellular activity of ChoK (**B**) in HepG2 and MCF7 cancer cell lines. Cells were incubated with the inhibitor for 48 h, and after its withdrawal, choline uptake and ChoKα activity were determined. Data represent the mean ± SEM of two independent experiments conducted in triplicate. Results are normalised to their respective controls. * *p* < 0.05, *** *p* < 0.0001.

**Figure 10 pharmaceutics-14-00426-f010:**
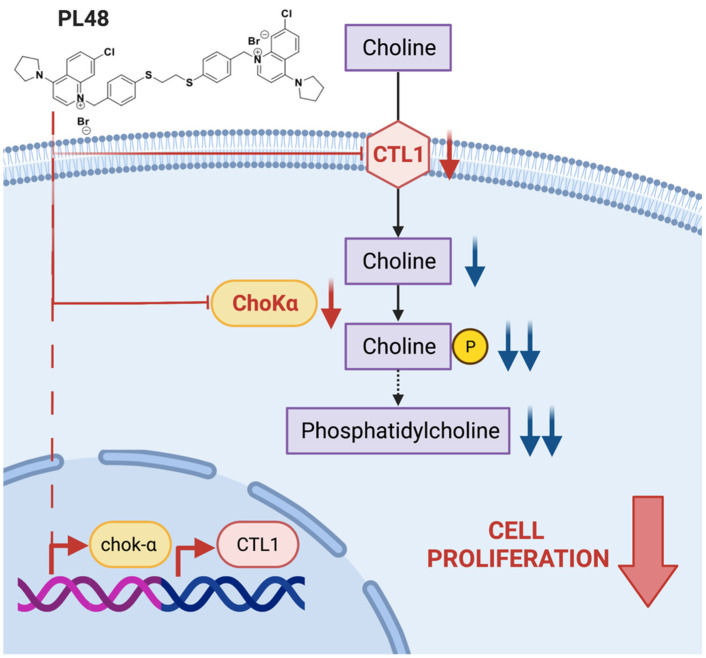
Scheme showing the action of PL48.

**Table 1 pharmaceutics-14-00426-t001:** Effect of PL48 on [methyl-^14^C]choline incorporation in HepG2 and MCF7. Cells were treated with 1 μM PL48 for 48 h. Then, they were incubated with [methyl-^14^C]choline as described in the methods section. The total incorporation of choline and the incorporation of choline into ChoP and PC are expressed as nmol of choline incorporated per mg of cell protein and represent the mean ± SEM of two independent experiments conducted in triplicate. * *p* < 0.05.

	HepG2			MCF7		
	Total	ChoP	PC	Total	ChoP	PC
**Control**	32.11 ± 2.09	23.74 ± 1.56	7.15 ± 0.61	13.46 ± 0.99	9.64 ± 0.62	2.21 ± 0.62
**1 μM PL48**	20.89 ± 1.33 *	16.3 ± 1.49 *	3.37 ± 0.30 *	10.22 ± 0.41 *	6.70 ± 0.63 *	2.38 ± 0.23

## Data Availability

Not applicable.

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
