# Peer review of "Anticancer Activity of the Choline Kinase Inhibitor PL48 Is Due to Selective Disruption of Choline Metabolism and Transport Systems in Cancer Cell Lines"

_pharmaceutics, 2022, doi:10.3390/pharmaceutics14020426_

Round 1
Reviewer 1 Report
This is an excellent work on a novel inhibitor of CHOK and certainly merits publication. Especially interesting is the incorporation of information about the expression and activity of choline transporters. However, the writing should be improved a great deal to make it easier to understand. On the one hand, more clarification is needed to truly understand the impact of this work. On the other hand, this manuscript would also benefit from proofreading and editing by a native English speaker.
Some general ideas: in each experiment, start by explaining the rationale, then tell the result, then refer back to the rationale to intepret the meaning of the result.
Introduction
"No PAINS (Pan Assay Interference 106
Structures) were detected for the new inhibitor, excluding the possibility of side effects 
due to the toxicity of chemical entities when interacting in the cellular medium." 106-108
When interacting with what? Are you claiming that it is not possible for this inhibitor to interact with any other component of the cell? Explain
Also, in the introduction, the authors should elaborate a little more on what past inhibitors: you say that "Our goal is to obtain more potent and selective inhibitors of 99
the enzyme that feature increased antiproliferative activity in cell lines of different ori- 100
gins." 
Good, why werent previous inhibitors potent and selective enough? Because of their shorter linker sequences? They werent lipophilotic enough? They had two many interference structures? The introduction should include a summary of the following:
1)the successes of past inhibitors. Ex. the first is HC-3
2) The limitations of past inhibitors. Ex. HC-3 is selective but has a high KD.
3) Why were these past inhibitors limited in value (Ex. they had short linkers)
4) Our novel inhibitor overcomes past limitations structurally because it has the following qualities ....(longer linkers? leading to higher antiproliferative activity, more selective?)
What is the origin of this inhibitor. Was it designed in the past. If so make clear the reference that indicates this. Otherwise, describe how the inhibitor was designed.
Results
"The Eadie–Hofstee plots show a concave appearance with two straight lines,"
You then refer to 2B and 2C, but there are three different plots. What is the difference between the plots? What does it mean to say that there is a concave appearence with two straight lines. Presumably the first plot is concave and the 2nd and 3rd are straight. Please explain further the difference between the 3 plots and how you interpet them.
Please show some indication of variability (error bars) in all plots.
"The inhibitor of the OCT family of transporters, TEA,"
Here you suddenly make the leap to OCTs (what are they) and their inhibitor TEA. Please first explain why it is important to test TEA."
"OCT-327 mediated choline uptake in HepG2 and MCF7 must be negligible"
This is confusing. Is OCT mediated uptake prevalent in other types of cells? If OCT were already reported not to be expressed in these cell types, what was the rationale for testing TEA?
"study confirms the functional role of CTL1"
I think at this point in the study you have have NOT confirmed it. Rather you have data that is consistent with the idea. Again you have to explain better why experiments with HC-3 were warranted.
"Lines 355-361". This looks like info that belongs in the introduction.
"As shown in Figure 386
5, PL48 inhibits ChoKα1 activity with an IC50 value of 0.658 ± 0.067 μM, well below that of 387
previously studied symmetrical biscationic compounds [25]."
A simple table showing the IC50 present inhibitor compared to the other inhibitors would be good here.
"Effect of PL48 on choline uptake in HepG2 and MCF7 cell lines 397" 
Which transporters are affected, do you think? CLT-1?
In Figure 7, which western blot refers to which type of cells?
"Figure 9. Scheme showing the action of PL48."
This is a very good graphic that sums up your results. However as you mention, PL48 both inhibits activity and expression of ChoK. Is there a way to show BOTH of these effects in the graphic?
Also, do you think you are also impacting CLT-1 activity? If not, how would you clarify this.
Is the antiproliferative effect more due to CLT-1 activity or CLT-1 expression, or more due to ChoK inhibition, expression?
How would you distinguish between these effects?
Conclusion.
We know that ChoK is a target. How can we be sure that CLT-1 is too?
Author Response
Thank you very much for your comments concerning our article entitled, “Anticancer activity of the choline kinase inhibitor PL48 is due to selective disruption of choline metabolism and transport systems in cancer cell lines” (pharmaceutics-1534093).
We have carefully read the comments and are now sending you a new version of our manuscript amended according to the points raised in the revision.
The manuscript has undergone English language edition by MDPI.
We hope that with these revisions to our paper you might now find it suitable for publication in Pharmaceutics.

Reviewer 2 Report
The authors presented a new compound PL48 as a bioisotere of EB-3P for inhibition of both choline uptake and ChoKα1 activity, and they evaluated its cellular cytotoxicity as well. Sulpher atom can be engaged in more interactions than Oxygen. The article is acceptable if the authors address some issues:
1- Describe PL48 chemical synthesis and characterization as it is a new compound. It may be similar to your previous work Scientific Reports | 6:23793 | DOI: 10.1038/srep23793.
2- The authors stated that sulphur free-electron pairs allow the formation of new hydrogen bonds within the choline-binding pocket of ChoKα1, improving its affinity for the enzyme. However, This claim should be supported by a molecular docking study between PL48 and ChoKα1 crystal structure which is available at protein data bank.
3- Support your idea with examples from literature where thionation afforded more bioactive agents than their Oxygen containing analogues.
Author Response

(The authors gave the same response as above.)

Reviewer 3 Report
The authors evaluate the antitumor activity and selectivity of a ChoK inhibitor PL48 in this work.
I have major concerns regarding methodology:
First of all, all the readouts evaluated (cell proliferation, enzyme activities, choline uptake, WB….) in cell lines (HepG2, MCF7) using PL48 could be recapitulated by cell toxicity (lysis, necrosis) induced by PL48. Cell damage also leads to less cell in proliferation test, less proteins in WB, les enzyme activities as well as less choline uptake). I will therefore analyse cell viability by the Lactate Dehydrogenase Assay for instance or other viability assay to evacuate this concern. Cell viability should be done in time dependant manner (6 hours, 24 hours…). Furthermore, I will assess apoptosis (caspase 3, Annexin V …..).
Another Major concern is about model of cell lines for instance MFC7 at basal state express very low level of CTL1, so any conclusion about effect of PL48 on CTL1 is unreliable. Following on from that the quality of data from figure 7 is quite mediocre, I will not trust this results, the same amount of protein should be loaded in the WB ( RIPA lysis, BCA protein quantification…).
If the results are still the same, how do you explain the fact that PL48 effects are quite similar is cell do express low level of CTL1 (MFC7) as well as cell with low ChoK expression (HepG2)? Add this in discussion.
Other revisions: line 21 (Gaining a…..) to 24 (… metabolic pathways) quite generic , that’s the aim of every researcher. I will focus on the specific aim of the paper (introduction on CTL1, ChoK, Choline pathway in cancer progression, stage disease, resistance to treatment, type of cancer….).
Fig 7, which ChoK is display in WB, total ChoK? Alfa isoform? Add the supplier reference antibodies “2.8 immunoblotting assay”
It’s take a while to reach on data and discussion part. Methods are quite standard such as WB, choline uptake assays … to be shrink.
Author Response

(The authors gave the same response as above.)

Round 2
Reviewer 2 Report
Thanks for your explanation regarding synthetic and docking parts.
Reviewer 3 Report
Substantial improvement of the manuscript has been achieved (additional experiments, controls, methodology, introduction as well data presentation).
I don't have access to a clean version
Round 3
Reviewer 3 Report
Clean version approved